# Peripheral T-Cell Lymphoma Possibly Due to Unrecognized Celiac Disease in an Elderly Patient: A Case Report

**DOI:** 10.3390/medicina57050471

**Published:** 2021-05-11

**Authors:** Anna Braszak-Cymerman, Marta K. Walczak, Katarzyna Iwanik, Paweł Kurzawa, Wiesław Bryl

**Affiliations:** 1Department of Internal Diseases, Metabolic Disorders and Hypertension, Poznań University of Medical Sciences, Szamarzewskiego 84, 60-569 Poznań, Poland; walczak_marta@interia.pl (M.K.W.); wieslawbryl@wp.pl (W.B.); 2 Department of Clinical Pathology, Poznań University of Medical Sciences, Przybyszewskiego 49, 60-355 Poznań, Poland; kiwanik@wp.pl (K.I.); pawel.kurzawa@skpp.edu.pl (P.K.); 3Department of Oncological Pathology, University Hospital of Lord’s Transfiguration, Partner of Karol Marcinkowski University of Medical Sciences, Szamarzewskiego 84, 60-569 Poznań, Poland

**Keywords:** celiac disease, lymphoma, elderly people, fever of unknown origin, delayed diagnosis

## Abstract

Celiac disease (CD) is widely perceived as a childhood disorder. However, it has been demonstrated that 19–34% of new CD cases are diagnosed in patients over 60 years of age and lack the typical presentation. A 76-year-old female was admitted to the clinic due to a recurrent fever that had lasted over a year accompanied by progressive weakness, weight loss of about 10 kg, dehydration, and malnutrition. The patient had undergone resection of a fragment of the small intestine due to perforation and abscess 13 years previously (at which time no histopathological examinations were performed). During the current hospitalization, despite extensive laboratory, microbiological, and imaging tests, no specific diagnosis was made. Symptomatic treatment and empirical antibiotic therapy were conducted, but the patient died on the twenty-seventh day of hospitalization due to progressive respiratory failure. The autopsy revealed peripheral T-cell lymphoma in the mesentery of the small intestine, uterus, cecum, lung, and mediastinal lymph nodes. Based on the clinical picture, we believe that the lymphoma was induced by long-term, undiagnosed CD. Current knowledge allows us to see age-related differences in the manifestation of celiac disease and to be alert to the possible late-stage complications of the disease. The lack of awareness of how CD’s symptoms vary with age may lead to misdiagnosis and serious consequences of delayed diagnosis, including death.

## 1. Introduction

Celiac disease (CD) is widely thought of as a childhood disorder. The typical case is often imagined to involve a pediatric patient with persistent diarrhea, abdominal pain, and malnutrition. However, recent studies have demonstrated that 19–34% of new CD cases are diagnosed in patients over 60 years of age [1]. Current knowledge also allows us to see age-related differences in the manifestation of CD [1,2]. The lack of awareness of how its various symptoms vary with age may lead to misdiagnosis and to the serious consequences of delayed diagnosis, which include death.

It has been shown that 2–3% of patients with CD will develop intestinal lymphoma, of which 65% is of T-cell origin. These lymphomas are believed to be one of the leading causes of death in patients with CD diagnosed in adulthood [3]. The risk of malignancies increases in older celiac patients and is associated with both the duration of gluten exposure and age. At a much higher risk of developing lymphoma are patients with refractory celiac disease (RCD) type 2. It has been found that this condition is associated with a high risk of developing enteropathy-associated T-cell lymphoma (EATL) [4].

## 2. Case Presentation

A 76-year-old female was admitted to the clinic due to a recurrent fever that had lasted over a year, accompanied by strong chills, progressive weakness, weight loss of about 10 kg, dehydration, malnutrition, and deteriorating general condition. The patient complained of recurrent urinary tract infections and 13 years previously had undergone resection of a fragment of the small intestine due to perforation and abscess (following recognition of malabsorption syndrome and cachexia; no histopathological examinations were performed, and no cause of the abscess or perforation was found). Three months prior to admission, the patient had been operated on for an enterovesical fistula. Histopathological examination of the material removed during that operation showed inflammatory changes in the intestinal wall. The patient had also reported periodic abdominal pain, recurrent diarrhea, and was diagnosed with osteoporosis with confirmed osteoporotic fractures of the spine.

Upon clinical examination, the patient was fully conscious and stable; cachexia and dehydration were noted. Baseline laboratory test results revealed anemia of chronic disease (with significantly elevated ferritin level), hypocalcemia, hypomagnesemia, vitamin D deficiency, and hypoalbuminemia. Laboratory tests also showed significantly elevated alkaline phosphatase, γ-glutamyltransferase, lactate dehydrogenase, aspartate transaminase, D-dimers, brain natriuretic peptide, and C-reactive protein. Microbiological tests were performed on blood samples, urine specimens, and wound swabs, but showed no abnormalities.

Abdominal, pelvic, and chest computed tomography (CT) scans confirmed a recanalized ureter with leakage of urine into the abdominal wall and patterns characteristic of pneumonia (Figure 1A). The abdominal CT scan also revealed a lymph node 20 × 12 mm to the left of the aorta, a small spleen (56 mm long), an increased density of periaortic fat tissue, and free fluid on the border of the abdominal cavity and pelvis—all described as postoperative, inflammatory, or cardiac-failure-related changes (Figure 1B).

The patient was catheterized and empirical broad-spectrum antibiotic therapy was begun, but a CT scan revealed the progression of the previously described changes. An 18F-fludeoxyglucose positron emission tomography (FDG–PET) study was scheduled.

Unfortunately, the patient died on the twenty-seventh day of hospitalization due to progressive respiratory failure. The specimens from the bladder fistula operation were histopathologically re-evaluated, revealing a malignant lymphoma of the small intestine (Figure 2A–D).

The autopsy also showed the presence of lymphoma in the mesentery of the small intestine, uterus, cecum, lung, and mediastinal lymph nodes. The lymphoma was CD45, CD3, CD7, CD43, CD4, CD30, and Ki-67 positive. The first histopathological study described enteropathy-associated T-cell lymphoma (EATL) according to the WHO classification. Based on further histopathological consultations, the lymphoma was classified as peripheral T-cell lymphoma not otherwise specified (PTCL-NOS). (Figure 3).

## 3. Discussion

We believe that the lymphoma was induced by long-term, undiagnosed CD. Despite growing knowledge and prevalence of CD, it remains heavily underdiagnosed, especially among older adults. In adults, the presentation of the disease is less marked than in children, and typical symptoms such as chronic diarrhea, weight loss, unresponsive iron deficiency anemia, and abdominal pain of unknown origin occur in less than 25% of cases [2]. More common are extraintestinal symptoms include anemia (often with high levels of ferritin and erythrocyte sedimentation rate), severe malnutrition, profound nutritional deficiency, accelerated osteoporosis, and osteomalacia. Malnutrition also leads to hypocalcemia and hypomagnesemia. Around 20% of celiac patients have abnormal liver function test results, caused by hepatocellular changes, a condition called celiac hepatitis [1]. Occasionally CD may present as mesenteric cavitation and spleen atrophy [5]. It is especially interesting here that gastrointestinal symptoms are not strongly marked [1,2].

As the age of diagnosis increases, antibody titers decrease and histological damage is less marked. It is common to find adults without villous atrophy showing only an inflammatory pattern in duodenal mucosa biopsies. The disease’s lower clinical, analytical, and histological expression in adults makes its diagnosis more complex than in children [2].

It has been shown that 2–3% of patients with CD will develop intestinal lymphoma, of which 65% is of T-cell origin. Peripheral T-cell lymphoma (PTCL) is a group of different T-cell lymphomas which develop in lymphoid tissues outside the bone marrow, such as the lymph nodes, spleen, gastrointestinal tract, and skin. The subtypes include PTCL-NOS, angioimmunoblastic T-cell lymphoma (AITL), anaplastic large cell lymphoma (ALCL), enteropathy-type T-cell lymphoma, and extranodal NK-T-cell lymphoma. In the intestines, EATL can be found, which is extremely rare and aggressive. It was previously recognized in two forms: Type 1, preceded by celiac disease, and Type 2, not linked to celiac disease, now known as monomorphic epitheliotropic intestinal T-cell lymphoma (MEITL). PTCL-NOS is the most common subtype of PTCL. It refers to a group of diseases that do not fit into any of the other PTCL subtypes.

Intestinal lymphomas are believed to be one of the leading causes of death in patients with CD diagnosed in adulthood [3]. The incidence of lymphoma is much greater in the sixth, seventh, and eighth decades of life. The rate of incidence of lymphomas is higher among older women over the age of 65 than among men of the same age [1]. The risk of malignancies increases in older celiac patients and is associated with both the duration of gluten exposure and age. Lymphoma is believed to be a particular complication of older CD, and people over the age of 50 newly diagnosed with CD should be monitored more closely, as they are more likely to develop lymphoma (in as many as one in ten cases) [6]. The increased risk of lymphoproliferative malignancy is also associated with a lack of mucosal healing. A higher incidence of lymphomas among patients with CD was found in patients with persistent villous atrophy [7]. Persistent villous atrophy for more than 6–12 months in celiac patients despite a strict gluten-free diet is classified as refractory celiac disease (RCD). RCD promotes the formation of lymphomas. Type 2 RCD can be defined as a low-grade intraepithelial lymphoma [4]. Approximately 50–60% of patients with Type 2 RCD will develop EATL within five years [8]. EATL can also develop in some patients who previously responded well to a gluten-free diet and later became symptomatic by developing RCD [9]. Some patients may develop acute complications of lymphomas, such as obstruction or perforation. In their retrospective study of patients without confirmed CD, Johnston et al. showed that perforation was more common in patients with small bowel lymphoma than in those with identified adenocarcinoma. Especially interesting was their finding that over half of patients with T-cell lymphomas had distant villous atrophy, suggesting that small bowel lymphoma is significantly associated with unrecognized CD [10]. This shows that celiac disease remains largely underdiagnosed, and the first symptom of CD may be its serious complications. Kotchetkov et al. also reported the case of a patient in whom EATL was the first manifestation of previously undiagnosed celiac disease [11].

Lymphomas generally follow a highly aggressive clinical course and have a poor prognosis [9]. EATLs can also present in disseminated forms with the involvement of mesenteric lymph nodes, the liver, spleen, lungs, and skin [3]. EATLs should be suspected in patients with elevated LDH and β2microglobulin, anemia, a BMI of around 18, and low serum albumin levels [12].

There are doubts about the diagnostic path when commonly available lab tests and imaging methods do not help to determine the diagnosis. The most valuable in such situations would seem to be the FDG-PET test, which will allow the detection of underlying malignancies, including difficult-to-diagnose lymphoma [13].

Unfortunately, due to the lack of a typical presentation of celiac disease and the rapidly deteriorating condition of our patient, a standard diagnosis for celiac disease was not performed. Lymphoma was also not suspected on the basis of laboratory and imaging tests made during hospitalization. Only the diagnosis of lymphoma at autopsy made it possible to discern a possible relationship between lymphoma and the atypical presentation of celiac disease in older age.

In an in-depth interview with the patient’s family, it turned out that in the past the patient followed a gluten-free diet that limited her diarrhea. After the episode of perforation, our patient did not agree to extend diagnostics with an endoscopic examination. Unfortunately, neither the improvement after the diet nor the earlier perforation of the small intestine and cachexia led to the suspicion of celiac disease.

Despite the fact that CD could not be confirmed by histopathological examination, and despite the lack of serological tests, on the basis of the overall clinical picture and laboratory tests, we suspect that the cause of the deteriorating general condition, osteoporosis and, ultimately, lymphoma, was undiagnosed CD.

The question remains whether the diagnosis of CD and lymphoma could have been made earlier. There are cases in which the perforation of the small intestine in patients with CD is the first symptom of an ongoing neoplasm [14]. It is possible that extending the diagnosis after perforation in our patient would have allowed the diagnosis of celiac disease or lymphoma at a much earlier stage of the disease. In such a situation, thorough histopathological and endoscopic diagnostics with laboratory tests for malabsorption disorders, including celiac disease, should be performed, and in the absence of a diagnosis, a PET-CT. Apart from surgical treatment, an acute abdomen accompanied by malnutrition should be given an accurate clinical and histopathological investigation to determine the underlying disease.

## 4. Conclusions

The case described here shows how serious, and difficult to diagnose, are complications of undiagnosed CD. It also indicates the importance of knowledge of the different manifestations of the disease depending on age.

## Figures and Tables

**Figure 1 medicina-57-00471-f001:**
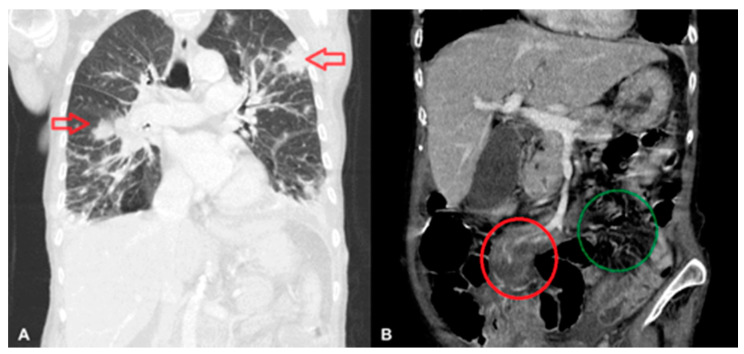
Lymphomatous infiltration of the lungs (indicated by arrows) (**A**), and of the mesentery of the small intestine (in the red circle); the right mesentery is shown in the green circle (**B**).

**Figure 2 medicina-57-00471-f002:**
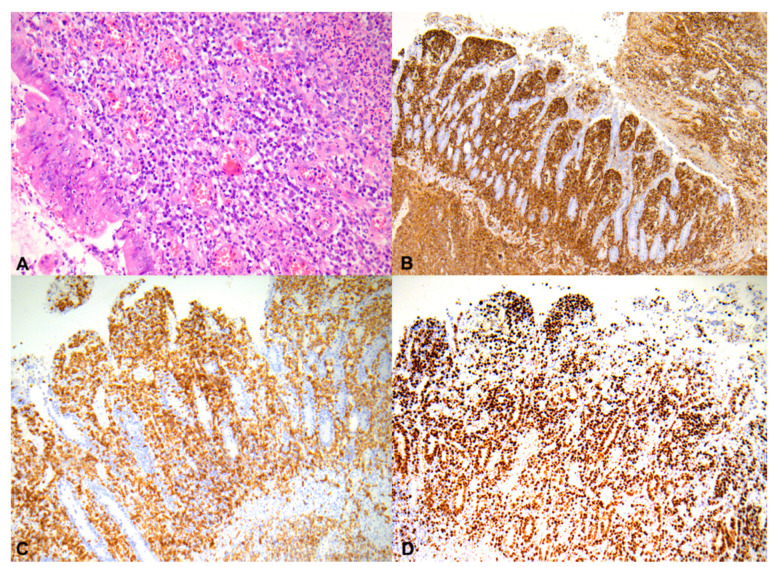
Neoplastic cells infiltrate the entire thickness of the small intestine, H&E, 200× (**A**); these neoplastic cells are strongly and diffusely positive for CD45, 50× (**B**); neoplastic lymphocyte showing dimmed, diffuse nuclear staining for CD3, 100× (**C**); tumor cells showing a high proliferation index (Ki-67 < 70%), 100× (**D**).

**Figure 3 medicina-57-00471-f003:**
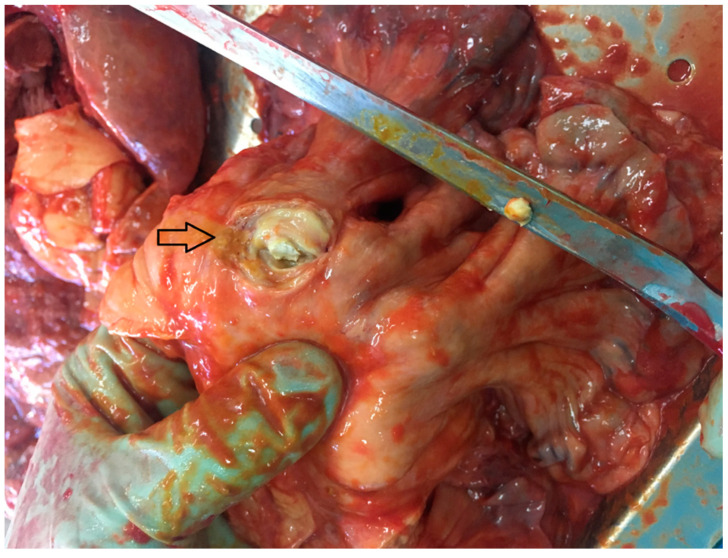
Tumor in the mesentery of the small intestine (indicated by the arrow).

## Data Availability

The patient data and images, as well as the written informed consent of the patient’s daughter, are available from the corresponding author on request.

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
