# Peer review of "Peripheral T-Cell Lymphoma Possibly Due to Unrecognized Celiac Disease in an Elderly Patient: A Case Report"

_medicina, 2021, doi:10.3390/medicina57050471_

Round 1

Reviewer 1 Report

Here are my comments regarding the case report titled "Peripheral T-cell lymphoma possibly due to unrecognized celiac disease in an elderly patient: A case report." by Braszak-Cymerman et al. 

  • The authors declare in the first line of discussion that they strongly believe that this peripheral t-cell lymphoma was caused by celiac disease. However, they do not present any evidence of celiac disease in the whole report. Celiac disease antibodies or duodenal biopsy was not taken. HLA DQ2/DQ8 status or family history of celiac disease is not presented. 
  • Another problem is the histopathological diagnosis of the lymphoma. Peripheral t-cell lymphoma (PTCL) is a group of the lymphomas and one of it's sublasses, the enteropathy associated t-cell lymphoma (EATL) type 1, is a result of refractory celiac disease type 2. EATL accounts for 5-8% of all PTCLs. The authors demonstrate that the lymphoma was CD3, CD45 and Ki-67 positive. However, important stainings for EATL are not mentioned: CD8, CD56, CD30, CD4, CD5 neither the TCR expression or genetic analyses of the lymphoma. In addition, EATL has specific histopathological properties, like pleomorphic medium to large sized tumour cells, eosinophils, histiocytes and necrosis. According to the presented analyses, the lymphoma of the patient (PTCL-NOS) does not indicate celiac disease as a backrgound disease.
  • In the discussion, the authors refer that celiac disease patients develop lymphoma. However, only a minor subset of celiac disease patients, the refractory celiac disease patients, develop commonly lymphoma. This should be corrected.
  • The authors cite an article from 1982 which is odd. We have lots of newer data.

Reviewer 2 Report

Summary

This case report describes a 76-year-old female patient admitted due to a year-long recurrent fever and progressive weakness, weight loss, dehydration and malnutrition. Unfortunately, the patient died on the 27th day of hospitalization, after which a lymphoma was discovered and classified as a peripheral T-cell lymphoma not otherwise specified (PTCL-NOS). The authors argues that this lymphoma was induced by a long-term, undiagnosed celiac disease (CD).

Broad comments

The authors have presented an interesting case regarding a patient suffering from a lymphoma and possibly CD, and highlights the complexity and possible severity of untreated CD. The text is, for most parts, easy to read and understand, but some sections need improvement.

My biggest issue with this paper is that you have not been able to confirm a CD diagnosis, neither by histopathological examination nor by serological tests. Were these not performed or did they fail to confirm CD? As it is now, the CD diagnosis is speculation, and it being the cause of the lymphoma even more so. You could very well be right, several of the symptoms do correspond to CD and investigating this further would be warranted as there is several publications regarding the association between CD and lymphoma, but I believe it cannot be ruled out that the symptoms seen could be due to the lymphoma and/or other causes as well. If you would have confirmed CD, then you could speculate if it induced the lymphoma, but even then, that would still be speculation as you only have one case.

My second biggest issue is what this paper adds in the understanding of the association with CD and lymphoma. There are several publications regarding this association, among those referenced by the authors. These publications seem to cover more patients and have stronger results compared to this paper. What specifically does this case report add in the knowledge and understanding of this association? That is not clear as it is now.

I think you also could discuss the literature some more. What are the similarities and differences of other reported cases compared to your patient? Since you do not have that much to back up your claims, I think it is even more important to compare with other findings. E.g. how likely is it that a similar lymphoma, without CD, would give similar symptoms and results as you have?

The resection 13 years ago – no cause for the abscess or perforation was found – do you have information on what type of examinations/tests/etc that were done to find/exclude possible causes? It might be interesting to add if you have any additional information. Could CD or the lymphoma be the cause already then, or were those ruled out at that point?

If this paper is to be published, I suggest that you make it absolutely clear that there is no confirmed CD diagnosis here and that you only hypothesize that CD might have been the cause for the lymphoma. Also, how your findings contribute to the field must be better described.

Specific comments

Line 17-27: Abstract – could be improved.

Line 17-19: First there is one sentence about the patient’s previous surgery. The next sentence I assume relates to the “present” day, but it could be interpreted as relating to the condition 13 years ago.

Line 24-27: These sentences are broad, covering implications and conclusions of your findings, and the previous sentences cover a description of the patient. I am lacking information on your findings in this case regarding CD.

Line 34-35: “Current knowledge also allows us to see age-related differences in the manifestation of CD” – so what is the current knowledge? I suggest that you write a little more about this, or at the very least add some references to where the reader can access this knowledge.

Line 89-90: To avoid confusion with lymphoma and CD, I suggest that you in this sentence write “CD” instead of “this disease”.

Line 90-91: One sentence about lymphoma, but the rest of this section is about CD. Consider moving this sentence or clarify this part. When I read it the first time I was not sure whether you were writing about lymphoma or CD.

Line 136-138: Confusing sentence – clarify.

Line 140-141: Confusing sentence – clarify.

Reviewer 3 Report

The authors present the case of a well-documented T-cell lymphoma with high-quality iconography. Note, however, that all the patient's symptoms and signs are well explained by the lymphoma itself (diarrhea, fever, weight loss, weakness, and malnutrition) and that the presence of osteoporosis at the age of 74 is common, even without evidence of malabsorption. The hypothesis that the patient had the underlying celiac disease is interesting but does not appear to be a sufficiently consistent basis and is merely speculative.

Unfortunately, there is a lack of clinical, serological, genetic (HLA-DQ2-DQ8 positive), or histopathological data that would have been relevant to support this hypothesis. The authors acknowledge that there was no histological analysis of the surgical specimen from the first intervention. On the other hand, the histological examination of the intestinal wall from the 2nd intervention only showed "inflammatory changes in the intestinal wall."

I think that the title of the manuscript is too emphatic. In the discussion, the authors should recognize more explicitly that the diagnosis of underlying celiac disease cannot be clearly supported.

However, the report has the merit of highlighting the importance that celiac disease continues to be an underdiagnosed disease and that this fact could have devastating consequences for the patient's future.

Round 2

Reviewer 1 Report

Here are my comments on the manuscript titled "Peripheral T-cell lymphoma possibly due to unrecognized celiac disease in an elderly patient: A case report." 

The revised version is better, however, the article suggest a link between PTCL-NOS and celiac disease which is misleading and this problem should be still corrected.

  • The title could be changed to "Lymphoma possibly due to...". As such it is misleading as PTCL-NOS is not linked to celiac disease according to literature and such data is neither reliably provided in the article.
  • In the abstract in the sentence "Based on the clinical picture, we strongly believe that the lymphoma was induced by long-term, undiagnosed CD." Please remove the word "strongly" as the supporting data is not strong. 
  • Please remove the word "strongly" also from the first sentence in Discussion and from the second to last chapter in discussion.
  • Introduction should have a some info about RCD and EATL, now it is unusually short.
  • It should be brought up to the unfamiliar reader that EATL is a subclass of PTCL-NOS.

Reviewer 2 Report

Line 144: Regarding the FDG-PET test – I assume you mean for help diagnosing lymphoma in this case? Could be clarified, to avoid misunderstandings.

Line 160-166: Interesting section. But it could perhaps be even better. In hindsight, what should have been done to successfully diagnose this patient earlier? What is the lesson learnt – what would you do differently if a new patient with similar symptoms comes to your clinic?  You write “…extending the diagnosis…”, but this is quite vague. As I wrote in the first review round, I had trouble finding how this paper contributes to the field, since the association between CD and lymphoma has been described (as you reference) in many papers before and with more data than you have. By extending this section a little maybe the value of this paper can be increased? 

Author Response

This manuscript is a resubmission of an earlier submission. The following is a list of the peer review reports and author responses from that submission.